# Dynamical and Coupling Structure of Pulse-Coupled Networks in Maximum Entropy Analysis

**DOI:** 10.3390/e21010076

**Published:** 2019-01-16

**Authors:** Zhi-Qin John Xu, Douglas Zhou, David Cai

**Affiliations:** 1NYUAD Institute, New York University Abu Dhabi, Abu Dhabi 129188, UAE; 2School of Mathematical Sciences, MOE-LSC and Institute of Natural Sciences, Shanghai Jiao Tong University, Shanghai 200240, China; 3Courant Institute of Mathematical Sciences and Center for Neural Science, New York University, New York, NY 10012, USA

**Keywords:** maximum entropy, pulse-coupled network, neural network, coupling structure, sparse coding

## Abstract

Maximum entropy principle (MEP) analysis with few non-zero effective interactions successfully characterizes the distribution of dynamical states of pulse-coupled networks in many fields, e.g., in neuroscience. To better understand the underlying mechanism, we found a relation between the dynamical structure, i.e., effective interactions in MEP analysis, and the anatomical coupling structure of pulse-coupled networks and it helps to understand how a sparse coupling structure could lead to a sparse coding by effective interactions. This relation quantitatively displays how the dynamical structure is closely related to the anatomical coupling structure.

## 1. Introduction

Binary-state networks—each node in one sampling time bin is binary-state—arise from many research fields, e.g., gene regulatory modeling and neural dynamics [1,2,3]. Statistical distributions of network states are essential to encode information [4,5,6,7,8]. For example, with statistical distributions of network states, experimental studies show that rats can perform awake replays of remote experiences in the hippocampus [9]. The number of network states of *n* binary-state nodes, $2^{n}$, exponentially grows as the node number, *n*, increases, which creates a challenge of characterizing the probability distribution of network states. Many works effectively characterize the distribution of network states in various systems, e.g., a network of ∼100 neurons [10], with a *low-order maximum entropy principle* (MEP) analysis [10,11,12,13,14,15,16,17,18]—a method with few (far less than $2^{n}$) non-zero effective interactions (see a precise definition in Equation (Equation 1)) constrained by low-order statistics. In the MEP analysis, the *dynamical structure* of the network, i.e., how nodes interact with each other in the recording of dynamical data, is characterized by effective interactions. This dynamical structure has been used to study the functional connectivity of networks [16,19]. For example, experimental studies show that the second-order effective interaction map of the retina is sparse and dominated by local overlapping effective interaction modules [19]. In this sense, those effective interactions can be regarded as a sparse coding of the information that encoded in the state distribution. Here, the sparseness is defined as the ratio of the number of non-zero effective interactions to $2^{n}$ (the number of total effective interactions). Several studies show that high-order effective interactions could be important for characterizing observed distributions of network states [6,10,20]. Although high-order effective interactions are required, the number of required non-zero high-order effective interactions are very small in these experimental studies [10]. What leads to the sparsity of effective interactions remains to be clarified.

We address how a sparse anatomical coupling structure (In the following, for simplicity, we use “coupling structure” instead of “anatomical coupling structure”.) could lead to a sparse coding by effective interactions. Network dynamical structure often closely relates to the underlying coupling structure [21]. For example, when the input of each node is independent from others, (i) high-order (≥2) effective interactions are zero in a network of no connections, and (ii) high-order effective interactions are large in a dense and strong connected excitatory network. To efficiently encode information, a realistic system often incorporates a coupling structure with certain features [22,23], e.g., sparsity, small-world, or scale-free. However, it is still unclear how the coupling structure affects the dynamical structure of effective interactions.

In this work, we consider a general class of pulse-coupled networks. The state of each node is binary-state, i.e., active when the node sends pulses to its child nodes, otherwise, silent. We first establish the connection between the coupling structure and the number of non-zero effective interactions in the full-order MEP analysis (constrained by all moments) through an observed *Fact*—which is independent from node dynamics. We then examine the observed Fact by numerical simulations. Through our analysis, we can estimate the number of non-zero effective interactions for a given coupling structure when the external input of each node is independent from one another. Our results show that a sparse network could lead to a lot of vanishing high-order effective interactions. For illustration, we verify our results by estimating the number of non-zero effective interactions for each order in a network with *Erdos–Renyi* connection structure, in which our estimated number is much smaller than $C_{n}^{k}$, the number of all possible *k*th-order effective interactions. Our results establish a connection between the dynamical structure and the network coupling structure. This connection provides an insight into how a sparse coupling structure can lead to a sparse coding scheme. In this work, we would mainly use neural networks as examples for illustration, while our results apply to general binary-state networks.

## 2. Results

In the following analysis, we use binary vector $V\left(l\right)=\left(\sigma_{1},\cdots ,\sigma_{n}\right)\in {\left\{0,1\right\}}^{n}$ to represent the state of *n* nodes within the sampling time bin labeled by *l*. To obtain correlations up to the *m*th-order requires to evaluate all ${\langle \sigma_{i_{1}}\cdots \sigma_{i_{M}}\rangle }_{E}$, where $1\leq i_{1}<i_{2}<\cdots <i_{M}\leq n$, 1≤M≤m, and ${\langle \cdot \rangle }_{E}$ is defined by ${\langle g(l)\rangle }_{E}=\sum_{l=1}^{N_{T}}g\left(l\right)/N_{T}$ for any function g(l) and $N_{T}$ is the total number of sampling time bins in the recording. The *m*th-order MEP analysis is to find the desired probability distribution P(V) for *n* nodes by maximizing the entropy $S\equiv -\sum_{V}P\left(V\right)\log P\left(V\right)$ subject to correlations up to the *m*th-order (m≤n). To solve this optimization problem [24], one introduces Lagrangian multipliers for each constraint, that is, $-J_{i_{1}\cdots i_{k}}$ for the constraint of ${\langle \sigma_{i_{1}}\cdots \sigma_{i_{k}}\rangle }_{E}$ and (Z−1) for the constraint of $\sum_{V}P\left(V\right)$ = 1. The optimization problem is
$$P_{m}\left(V\right)=\arg\max_{P(V)}-\sum_{V}P\left(V\right)\log P\left(V\right)-\sum_{k=1}^{m}\sum_{i_{1}<\cdots <i_{k}}^{n}F_{i_{1}\cdots i_{k}}-F_{0},$$
where
$$F_{0}=\left(Z-1\right)\left(\sum_{V}P\left(V\right)-1\right),$$
$$F_{i_{1}\cdots i_{k}}=-J_{i_{1}\cdots i_{k}}\left(\sum_{V}P\left(V\right)\prod_{j=1}^{k}\sigma_{i_{j}}\left(V\right)-{\langle \sigma_{i_{1}}\cdots \sigma_{i_{k}}\rangle }_{E}\right),$$
and $\sigma_{i_{j}}\left(V\right)$ is the state of the $i_{j}$th node in the network state *V*. Since the entropy *S* is a convex function of P(V), the unique distribution can be solved by taking a derivative of the above objective function with respect to each P(V) as follows: (1)$$P_{m}\left(V\right)=\frac{1}{Z}\exp\left(\sum_{k=1}^{m}\sum_{i_{1}<\cdots <i_{k}}^{n}J_{i_{1}\cdots i_{k}}\sigma_{i_{1}}\cdots \sigma_{i_{k}}\right),$$
where, following the terminology of statistical physics, we call $J_{i_{1}\cdots i_{k}}$ a *k*th-order effective interaction (1≤k≤m), the partition function *Z* is the normalization factor. Equation (Equation 1) is referred to as the *m*th-order MEP distribution. When m<n, the constants $J_{i_{1}\cdots i_{k}}$ can be determined by all constraints through a commonly used iteration method [13]. When m=n, all order moments of the experimentally observed distribution are used as constraints; thus, the above optimization problem has only one feasible solution, that is, the experimentally observed distribution [25,26].

First, we discuss the relationship between effective interactions and the statistical distribution of network states. By taking the logarithm of both sides of Equation (Equation 1) for $P_{n}\left(V\right)$, we can get a set of linear equations of all-order effective interactions for all states *V*. Since $P_{n}$ is the same as the experimentally observed distribution [25], we can obtain the effective interactions in $P_{n}$ in terms of the experimentally observed distribution [27]. For example, n=3, we can obtain $J_{1}=\log\left(P_{100}/P_{000}\right)$ and $J_{12}=\log\left(P_{110}/P_{010}\right)-J_{1}$, where $P_{\sigma_{1}\sigma_{2}\sigma_{3}}$ represents the probability of the network state $\left(\sigma_{1},\sigma_{2},\sigma_{3}\right)$. By applying $P\left(\sigma_{1},\sigma_{2},\sigma_{3}\right)=P\left(\sigma_{1}|\sigma_{2},\sigma_{3}\right)P\left(\sigma_{2},\sigma_{3}\right)$, we have $J_{1}=\log\frac{P(\sigma_{1}=1|\sigma_{2}=0,\sigma_{3}=0)}{P(\sigma_{1}=0|\sigma_{2}=0,\sigma_{3}=0)}$ and $J_{12}=\log\frac{P(\sigma_{1}=1|\sigma_{2}=1,\sigma_{3}=0)}{P(\sigma_{1}=0|\sigma_{2}=1,\sigma_{3}=0)}-J_{1}≜J_{1}^{1}-J_{1}$. Our earlier study has shown a recursive structure among effective interactions, that is, the (k+1)st-order effective interaction $J_{123\cdots (k+1)}$ can be obtained as follows [27]: first, we switch the state of the (k+1)st node in $J_{123\cdots k}$ from silent to active to obtain a new term $J_{123\cdots k}^{1}$, e.g., from $J_{1}$ to $J_{1}^{1}$; then, we subtract $J_{123\cdots k}$ from the new term to obtain $J_{123\cdots (k+1)}$, i.e.,
(2)$$J_{123\cdots (k+1)}=J_{123\cdots k}^{1}-J_{123\cdots k}.$$

Without loss of generality, we randomly select two nodes labeled by 1 and 2. By the recursive relation, any *k*th-order effective interaction that includes node 1 and 2 can be expressed as the summation of terms with the following *basic form*:(3)$$\begin{array}{cc} J_{12}^{b}\left(\sigma_{3},\cdots ,\sigma_{n}\right) & =\log\frac{P(\sigma_{1}=1|\sigma_{2}=1,\sigma_{3},\cdots ,\sigma_{n})}{P(\sigma_{1}=0|\sigma_{2}=1,\sigma_{3},\cdots ,\sigma_{n})}\\ & \;-\log\frac{P(\sigma_{1}=1|\sigma_{2}=0,\sigma_{3},\cdots ,\sigma_{n})}{P(\sigma_{1}=0|\sigma_{2}=0,\sigma_{3},\cdots ,\sigma_{n})}. \end{array}$$

Note that we use superscript *b* to denote “*basic*”. For any *i* and *j* (i≠j), the states of node *l* (l≠i,j) in $J_{ij}$ is silent. $J_{ij}^{b}$ is a function of states of nodes $\sigma_{l}$’s (l≠i,j) by replacing the silent state of every node *l* (l≠i,j) in $J_{ij}$ with variable $\sigma_{l}\in \left\{0,1\right\}$. For example, $J_{123}=J_{12}^{b}\left(1,0,\cdots ,0\right)-J_{12}^{b}\left(0,0,\cdots ,0\right)$ and $J_{1234}=\left[J_{12}^{b}\left(1,1,0,\cdots ,0\right)-J_{12}^{b}\left(0,1,0,\cdots ,0\right)\right]-J_{123}$. We can observe that if nodes 1 and 2 are independent conditioned on all other nodes, i.e., $P\left(\sigma_{1}|\sigma_{2}=1,\sigma_{3},\cdots ,\sigma_{n}\right)=P\left(\sigma_{1}|\sigma_{2}=0,\sigma_{3},\cdots ,\sigma_{n}\right)$, any effective interaction containing these two nodes is zero.

Next, we would show what kind of coupling structure could entail the conditional independence of two nodes. Here, we define some notations. In any sampling time bin [0,Δ) with state $V=(\sigma_{1},\cdots ,\sigma_{n})$, ∀t∈[0,Δ), we denote $I_{i,t}$ as node *i*’s input from the outside of the network, denote $w_{ij}\left(t\right)$ as the input from the node *i* to node *j*, denote C(i) as the set of all child nodes of node *i*, denote $U_{i}=C\left(i\right)\cup \left\{i\right\}$, denote P(e) as the probability of event *e*, denote $U_{0}=\left\{1,2,\cdots ,n\right\}$.

**Fact** **1.**
*For n pulse-coupled nodes with binary-state dynamics on a network with a coupling structure $G_{0}$, in any sampling time bin [0,Δ), ∀t∈[0,Δ), $\forall i_{1},j_{1}\in U_{0}$, we assume that: (a) the external inputs of each node are independent from one another, i.e., $P\left(I_{i_{1},t},I_{j_{1},t}\right)=P\left(I_{i_{1},t}\right)P\left(I_{j_{1},t}\right)$; (b) whether a parent node sends spikes to its child nodes only depends on its state, i.e., $P\left(w_{i_{1}j_{1}}\left(t\right),V\right)=W\left(\sigma_{i_{1}},i_{1},j_{1},t\right)$, where W(·,·,·,·) is a real function. $\forall i,j\in U_{0}$, if they neither are connected nor share any common child node, i.e., $U_{i}\cap U_{j}=\phi$, then, node i and j are independent conditioned on the state of all other nodes, i.e.,*
(4)$$P\left(\sigma_{i},\sigma_{j}|H\right)=P\left(\sigma_{i}|H\right)P\left(\sigma_{j}|H\right),$$
*where H is a possible state of nodes in $U_{0}\setminus \left\{i,j\right\}$.*


We justify our two assumptions as follows: to avoid the influence of correlation in external inputs when we are studying the relation between the dynamical structure and the coupling structure, we assume that the external input of each node is independent from one another, i.e., assumption (a). The second assumption implicates a Markov-like property; that is, for a connected pair of pulse-coupled nodes in an equilibrium state, the pulse from the parent node to the child node only depends on the state of the parent node but is independent from inputs imposed on the parent node. For example, in neural networks, a neuron sends out spikes only when this neuron is active, regardless of what inputs are imposed on the neuron.

The argument for the conclusion in Equation (Equation 4) is as follows. By assumption (a), node *i* and node *j* can be dependent only through the coupling structure $G_{0}$. When we are considering how node *i* and node *j* affect each other by changing their states through the coupling structure $G_{0}$, we can consider a simplified coupling structure, $G_{1}$, which ignores those connections that are independent from states of node *i* and node *j*, i.e., $\sigma_{i}$ and $\sigma_{j}$. $\forall k\in U_{o}\setminus \left\{i,j\right\}$, i.e., any other node *k*, its state $\sigma_{k}$ is fixed when we are considering the conditional probability in Equation (Equation 4). By assumption (b), for node *k*’s any child node *l*, the input from node *k* to node *l* is independent from $\sigma_{i}$ and $\sigma_{j}$. Thus, the connections started from those nodes in $U_{o}\setminus \left\{i,j\right\}$ are fixed for different states of $\sigma_{i}$ and $\sigma_{j}$. Therefore, $G_{1}$ is a simplified coupling structure that only keeps those connections that originated from node *i* and node *j* in $G_{0}$. In $G_{1}$, any connection only exists in either sub-network $U_{i}$ or sub-network $U_{j}$. Under the condition $U_{i}\cap U_{j}=\phi$, i.e., they neither are connected nor share any common child node, sub-network $U_{i}$ and sub-network $U_{j}$ are two isolated sub-networks. $\sigma_{i}$ and $\sigma_{j}$ cannot affect each other by changing their states through the coupling structure $G_{1}$, that is, node *i* and *j* are independent conditioned on the states of all other nodes.

Figure 1 displays an example to illustrate our observed Fact. The coupling structure $G_{0}$ is shown in Figure 1a. We focus on node 1 and node 2, where they neither are connected nor share any child node. When the state of other nodes (black) are fixed, all outputs from black nodes can be ignored in the simplified coupling structure $G_{1}$, as shown in Figure 1b. Node 1 and node 2 respectively belong to two separate sub-networks. Therefore, nodes 1 and node 2 are independent conditioned on the state of all other nodes.

Based on the recursive structure of effective interactions and the observed Fact, we reach the following conclusion: with the two assumptions in the observed Fact, for a group of nodes $\left\{i_{1},i_{2},\cdots ,i_{k}\right\}$, if there exists at least one pair of nodes that neither are connected nor share any child node, effective interaction $J_{i_{1},i_{2},\cdots ,i_{k}}$ is zero.

The system we would use to examine our conclusion is an integrate-and-fire (I&F) network, a general pulse-coupled network, with both excitatory and inhibitory nodes [21]. For the *i*th node, the dynamics of its state variable $x_{i}$ with time scales τ are governed by
(5)$${\dot{x}}_{i}=-\frac{x_{i}}{\tau }-\left(g_{i}^{\mathrm{bg}}+g^{\mathrm{ex}}\right)\left(x_{i}-x_{\mathrm{ex}}\right)-g_{i}^{\mathrm{in}}\left(x_{i}-x_{\mathrm{in}}\right),$$
where $x_{\mathrm{ex}}$ and $x_{\mathrm{in}}$ are the reversal values of excitation (ex) and inhibition (in), respectively. $g_{i}^{\mathrm{bg}}=f\sum_{k}H\left(t-T_{i,k}^{F}\right)\exp\left[-\left(t-T_{i,k}^{F}\right)/\sigma^{\mathrm{ex}}\right]$ is the background input with magnitude *f* and time scale $\sigma^{\mathrm{ex}}$, $T_{i,k}^{F}$ is a Poisson process with rate μ, H(·) is the Heaviside function, $g_{i}^{\mathrm{ex}}=\sum_{j}\sum_{k}S_{ij}^{\mathrm{ex}}H\left(t-T_{j,k}^{\mathrm{ex}}\right)\exp\left[-\left(t-T_{j,k}^{\mathrm{ex}}\right)/\sigma^{\mathrm{ex}}\right]$ is the excitatory input from other *j*th excitatory nodes, and $g_{i}^{\mathrm{in}}=\sum_{j}\sum_{k}S_{ij}^{\mathrm{in}}H\left(t-T_{j,k}^{\mathrm{in}}\right)\exp\left[-\left(t-T_{j,k}^{\mathrm{in}}\right)/\sigma^{\mathrm{in}}\right]$ is the inhibitory input from other *j*th inhibitory nodes. The *j*th excitatory (inhibitory) node $x_{j}$ evolves continuously according to Equation (Equation 5) until it reaches a firing threshold $x_{\mathrm{th}}$. That moment in time is referred to as a firing event (say, the *k*th spike) and denoted by $T_{j,k}^{\mathrm{ex}}$ ($T_{j,k}^{\mathrm{in}}).$ Then, $x_{j}$ is reset to the reset value $x_{r}$ ($x_{\mathrm{in}}<x_{r}<x_{\mathrm{th}}<x_{\mathrm{ex}}$) and held $x_{r}$ for an absolute refractory period of $\tau_{\mathrm{ref}}.$ Each spike emerging from the *j*th excitatory (inhibitory) node causes an instantaneous increase $S_{ij}^{\mathrm{ex}}$ ($S_{ij}^{\mathrm{in}}$) in $g_{i}^{\mathrm{ex}}$ ($g_{i}^{\mathrm{in}}$), where $S_{ij}^{\mathrm{ex}}$ and $S_{ij}^{\mathrm{in}}$ are the excitatory and inhibitory coupling strengths, respectively. The model (Equation 5) describes a general class of physical networks [1,3,21,28,29]. To be intuitive, the I&F model described in Equation (Equation 5) can be understood through a resistor–capacitor circuit. Each neuron is a leaky circuit, which consists of a capacitor with dimensionless capacitance as 1 in parallel with two resistors. $x_{i}$ is the voltage. ${\dot{x}}_{i}$ on the left-hand side is the current which passes through the capacitor. On the right-hand side, $-x_{i}/\tau$ is the leaky current. The first resistor has a reversal potential $x_{ex}$ with conductance $\left(g_{i}^{\mathrm{bg}}+g^{\mathrm{ex}}\right)$. Therefore, the second term on the right-hand side obtained by the Ohm’s law is the current that passes through the first resistor. Note that the conductance is affected by inputs. It is similar for the third term on the right-hand side, i.e., the current that passes through the second resistor.

In the first example, two excitatory and two inhibitory I&F nodes form a ring coupling structure (Figure 2a). For any pair of nodes, say, node *i* and *j*, we compute $\Delta_{ij}\left(H\right)=\left|P\left(\sigma_{i}=1|\sigma_{j}=1,H\right)-P\left(\sigma_{i}=1|\sigma_{j}=0,H\right)\right|$, where *H* is a state vector of other two nodes. By our observed Fact, the conditional independent pairs are (neuron1,neuron3) and (neuron2,neuron4), and other pairs are categorized as dependent pairs. In Figure 2b, the strengths of $\Delta_{ij}\left(H\right)$ of independent pairs (green) are almost two orders of magnitude smaller than those of dependent pairs (red). We then shuffle spike trains of each node. We similarly compute $\Delta_{ij}\left(H\right)$ for 10 different pieces of shuffled data. Blue dots and cyan dots in Figure 2b are results of all shuffled data of dependent pairs and independent pairs, respectively. The strength of $\Delta_{ij}\left(H\right)$ of independent pairs (green)—computed from the observed data—are within the statistical error of shuffled data. We then solve effective interactions in the full-order MEP analysis $P_{n}$ for this ring network. As shown in Figure 2c, the effective interaction strengths of independent pairs ($J_{24}$ and $J_{13})$ are within the statistical error of shuffled results (red). Since every high-order (≥3) effective interaction includes at least one independent pair of nodes, as predicted, the strengths of all high-order effective interactions are within the statistical error of shuffled results as shown in Figure 2d.

In the second example in the second row in Figure 2, results are similar in that dependent pairs and independent pairs can be identified through our observed Fact, and the strength of any effective interaction that includes the independent pair of nodes (node 1 and node 3) is within the statistical error of shuffled data. In this example, as shown in Figure 2h, $J_{124}$ is very small, i.e., within the statistical error of shuffled results. However, $J_{124}$ does not include the independent pair of nodes (node 1 and node 3); thus, the theoretical estimation of zero-strength effective interactions misses $J_{124}$. This example indicates that our theoretical estimation result may give rise to an upper bound of the number of non-zero effective interactions. In contrast, for a network of all excitatory nodes with the same coupling structure as the one in Figure 1e, $J_{124}$ is significantly larger than zero (not shown). Since the estimation of the strength of high-order effective interactions involves the estimation of high dimensional joint probability distribution, a very long recording due to the curse of dimensionality constrains us from examining $\Delta_{ij}\left(H\right)$ for a large network.

Based on the relation between the coupling structure and effective interactions, the number of non-zero high-order effective interactions can be small in a sparsely connected network compared with $C_{n}^{k}$, which is the number of all possible *k*th-order interactions. For example, we estimate the number of each-order non-zero effective interactions in a network with an *Erdos–Renyi* connection structure. We generate 1000 Erdos-Renyi random networks of 100 nodes (the same connection probability but different random samples). In each network, for any pair of node *i* and node *j*, we assign the connection from node *i* to node *j* by the following rule: we generate a random number from the uniform distribution on [0,1]; if the number is smaller than 0.05, we assign a connection from node *i* to node *j*. As shown in Figure 3, the number of non-zero *k*th-order (k>1) effective interactions is much smaller than $C_{100}^{k}$ (too large to be shown). The number of high-order effective interactions (order higher than 11th) almost vanishes (order higher than 20th not shown). In this example, the sparseness, i.e., the ratio of the number of non-zero effective interactions to $2^{n}$, is less than $10^{-24}$.

## 3. Conclusions and Discussion

In summary, we have established a relation between effective interactions in MEP analysis and the anatomical coupling structure (“coupling structure” for simplicity in the following) of pulse-coupled networks to understand how a sparse coupling structure could lead to a sparse coding by effective interactions. For example, the sparseness of the case in Figure 3 is less than $10^{-24}$. Since effective interactions characterize how nodes interact with each other in the recording of dynamical data, i.e., network dynamical structure, our study quantitatively displays how the dynamical structure closely relates to the coupling structure.

Even though high-order effective interactions are often much smaller compared with low-order ones [27], it is still unclear why small high-order effective interactions do not accumulate to have a significant effect in a large network [10,30]. For example, it has been shown that MEP distribution with sparse low-order effective interactions—non-zero effective interactions are sparse and vanish when the order is higher than the eighth-order—can well capture the state distribution of 99 ganglion cells in the salamander retina responding to a natural movie clip or natural pixel [10]. In this study, we show that a large amount of effective interactions vanish in a sparse coupling structure, thus rationalizing the absence of the accumulation of high-order interactions for a large network.

Finally, we point out that some important issues remain to be elucidated in the future. First, we have ignored correlations in external inputs when estimating the number of non-zero effective interactions. Correlated inputs can induce non-zero high-order effective interactions [31]. It still needs to be considered how the statistics of inputs affect the sparsity of effective interactions. Second, current algorithms for estimating non-zero effective interactions (not limited to the second-order) for a large network (e.g., ∼100 nodes) are very slow, e.g., Monte Carlo based methods [30,32]. Our undergoing work is developing a fast algorithm that exploits the sparsity of effective interactions. We have seen an indication that the algorithm can work well for an I&F network with sparse coupling structure; however, that work has yet to be fully verified to be conclusive.

## Figures and Tables

**Figure 1 entropy-21-00076-f001:**
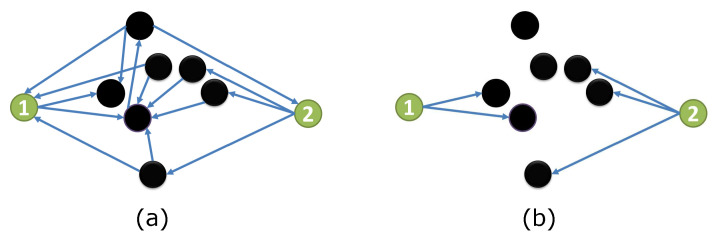
Structure vs. simplified structure.

**Figure 2 entropy-21-00076-f002:**
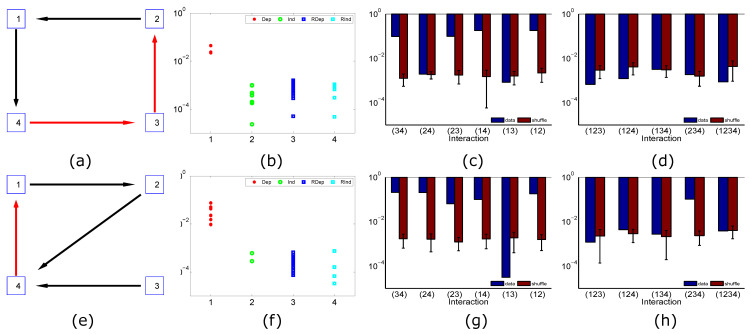
Anatomical structure vs. effective interactions of integrate-and-fire networks. Each row shows a numerical case. In the first column, black arrows and red arrows represent excitatory and inhibitory connections, respectively. In the second column, red and green dots are the strengths of $\Delta_{ij}\left(H\right)$ of dependent and independent pairs, respectively. Blue dots and cyan dots are the strengths of $\Delta_{ij}\left(H\right)$ of dependent and independent pairs from ten shuffled spike trains, respectively. Each dot is for one $\Delta_{ij}\left(H\right)$. The third and fourth columns display absolute effective interaction strengths (blue bars). The corresponding node indexes for each effective interaction are shown in the abscissa. The mean and standard deviation of absolute strengths of each effective interaction of ten shuffled spike trains are also displayed by garnet bars. The simulation time for each network is $1.2\times 10^{8}\;\mathrm{ms}$. The time bin size for analysis is 10ms [12,13]. Independent Poisson inputs for each network are $\mu =0.1\;{\mathrm{ms}}^{-1}$ and $f=0.1\;{\mathrm{ms}}^{-1}$. The firing rate of each node is about 50Hz. Parameters are chosen [28] as $x_{\mathrm{ex}}=14/3$, $x_{\mathrm{in}}=-2/3$, $\sigma^{\mathrm{ex}}=2\;\mathrm{ms}$, $\sigma^{\mathrm{in}}=5\;\mathrm{ms}$, τ=20ms, $x_{\mathrm{th}}=1$, $x_{r}=0$, and $\tau_{\mathrm{ref}}=2\;\mathrm{ms}$, $S_{ij}^{\mathrm{ex}}=S_{ij}^{\mathrm{in}}=0.02$.

**Figure 3 entropy-21-00076-f003:**
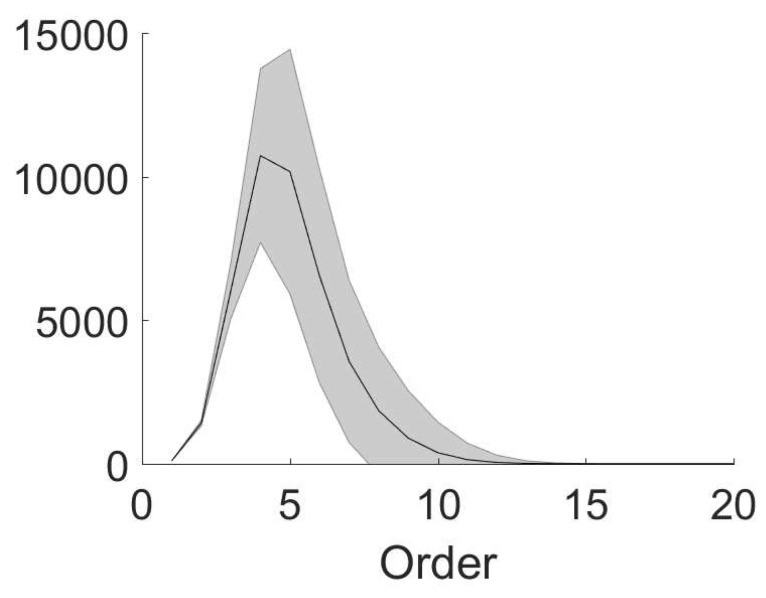
Non-zero effective interactions in the Erdos-Renyi random networks. We generate 1000 Erdos-Renyi random networks of 100 nodes (the same connection probability but different random samples). The connection probability between two nodes is 0.05. The number of non-zero effective interaction is plotted against effective interaction order. The mean and standard deviation are respectively shown by the black line and shaded area.

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
