# Peer review of "Dynamical and Coupling Structure of Pulse-Coupled Networks in Maximum Entropy Analysis"

_entropy, 2019, doi:10.3390/e21010076_

Round 1
Reviewer 1 Report
The manuscript provides a clear connection between the structure of coupling (between pulse interacting binary components of a system) and the dynamical structure of effective high-order interactions in Maximum Entropy Principle analysis. This connection rationalizes in a successful way the observed absence of accumulation of higher-order interactions for large sparse networks.
I recommend its publication, after some minor improvements on the description of the specific integrate and fire (I&F) model (equation 5 and following paragraph) used, for the benefit of those readers not familiar with I&F models.
Author Response
Dear reviewer,
We thank you for your thoughtful comments on our previous manuscript (ID: entropy-393338) and we have revised the manuscript accordingly. In the attached file, we detail our responses to your comments. We hope that the revised manuscript now satisfies your requirements and hereby resubmit the revised manuscript for publication in Entropy.
Sincerely yours,
Zhiqin John Xu

Reviewer 2 Report
The author's proposed methods to quantify the relationship between the dynamical structure of a neural network and the coupling structure that reflecting effective interaction between the nodes. Methods to quantify the property of the network dynamics and network structure is surely important, and that is consistent with the aim of the journal and is suitable for the interests of the readers of the journal.
However, a description of the proposed methods is not clear, and the evaluation of the proposed methods is not enough. The estimation of the high-order correlation is the key point of the paper, but the authors do not show the related result with the appropriate form. the manuscript is required major revision. If the authors conducted on the following the revision points, the manuscript will be acceptable.
In the results section, it is difficult for me to catch how the method work.
Regarding Eq. (1), please clarify how did you derive the Eq from the MEP analysis (maximizing the entropy).
In Eq. (3), please clarify what the “b” stands for.
The authors should define “sparse coding.”
In line 51, “child notes” might be a typo.
Line 107-109, the sentence of “J_{124} is very small…” does not make sense. Is this about Fig.2h? Please describe the properties of the higher-order statistics.
In the paragraph starting from line 114, please describe the way of the configuration of Erdos-Renyi connection.
In the discussion section, the authors used terminology “sparse coding by effective interactions,” “dynamical structure,” “coupling structure”, but the relation among these terms are ambiguous. Please define the meaning of them.
Author Response

(The authors gave the same response as above.)

Reviewer 3 Report
393338 20181110
This paper deals with dynamical and coupling structure of pulse-coupled networks in maximum entropy analysis. I would like to point out following as:
1. In introduction, this paper is ambiguous the object and subject. I hope authors should add more detailed and clear object of this paper, why this paper is necessary in the side of reader.
2. The description of previous work should add more detailed content for MEP in introduction.
3. I think main content or motivation of this paper is very weak. This authors should add more detailed contents including figure and block diagram to easily understand main contents.
Author Response

(The authors gave the same response as above.)

Round 2
Reviewer 3 Report
I Think this paper well revised according to reviewer's point out. Thus I would like to decide as an "Accept"